# Concurrent HPV DNA testing and a visual inspection method for cervical precancer screening: A practical approach from Battor, Ghana

Kofi Effah[1], Ethel Tekpor[1], Comfort Mawusi Wormenor[1], Joseph Emmanuel Amuah[1,2], Nana Owusu Essel[3]*, Bernard Hayford Atuguba[1], Gifty Belinda Klutsey[1], Edna Sesenu[1], Georgina Tay[1], Faustina Tibu[1], Seyram Kemawor[1], Isaac Gedzah[1], Esu Aku Catherine Morkli[1], Stephen Danyo[1], Patrick Kafui Akakpo[4]

1 Catholic Hospital, Battor, via Sogakope, Volta Region, Ghana, 2 School of Epidemiology and Public Health, Faculty of Medicine, University of Ottawa, Ottawa, ON, Canada, 3 Department of Emergency Medicine, College of Health Sciences, Faculty of Medicine and Dentistry, University of Alberta, Edmonton, AB, Canada, 4 Department of Pathology, University of Cape Coast, School of Medical Sciences, Clinical Teaching Center, Cape Coast, Ghana

* nanaowus@ualberta.ca

**Data Availability Statement:** All relevant data are within the paper, and relevant datasets can be

## Abstract

Cytology-based cervical cancer screening programs have been difficult to implement and scale up in developing countries. Thus, the World Health Organization recommends a 'see and treat' approach by way of hr-HPV testing and visual inspection. We aimed to evaluate concurrent HPV DNA testing and visual inspection in a real-world low-resource setting by comparing the detection rates of concurrent visual inspection with dilute acetic acid (VIA) or mobile colposcopy and hr-HPV DNA testing to standalone hr-HPV DNA testing (using the *care*HPV, GeneXpert, AmpFire, or MA-6000 platforms). We further compared their rates of loss to follow-up. This retrospective, descriptive cross-sectional study included all 4482 women subjected to cervical precancer screening at our facility between June 2016 and March 2022. The rates of EVA and VIA 'positivity' were 8.6% (95% CI, 6.7–10.6) and 2.1 (95% CI, 1.6–2.5), respectively, while the hr-HPV-positivity rate was 17.9% (95% CI, 16.7–19.0). Overall, 51 women in the entire cohort tested positive on both hr-HPV DNA testing and visual inspection (1.1%; 95% CI, 0.9–1.5), whereas a large majority of the women tested negative (3588/4482, 80.1%) for both and 2.1% (95% CI, 1.7–2.6) tested hr-HPV-negative but visual inspection 'positive'. In total, 191/275 (69.5%) participants who tested hr-HPV positive on any platform, as a standalone test for screening, returned for at least one follow-up visit. In light of factors such as poor socioeconomic circumstances, additional transportation costs associated with multiple screening visits, and lack of a reliable address system in many parts of Ghana, we posit that standalone HPV DNA testing with recall of hr-HPV positives will be tedious for a national cervical cancer prevention program. Our preliminary data show that concurrent testing (hr-HPV DNA testing alongside visual inspection by way of VIA or mobile colposcopy) may be more cost-effective than recalling hr-HPV-positive women for colposcopy.

accessed at: https://doi.org/10.6084/m9.figshare.21801268.v2.

**Funding:** The authors received no specific funding for this work.

**Competing interests:** The authors have declared that no competing interests exist.

# Introduction

Globally, about half a million women develop cervical cancer annually [1] with over 85% of cases occurring in developing countries, making it the second most frequent cause of cancer deaths [2]. Cervical precancer screening remains a key strategy in reducing the global disease burden. In developing countries, however, infrastructural challenges, lack of trained personnel, high costs, and long waiting times for test results make cytological screening ill-suited [3]. To achieve comprehensive cervical cancer control, human papillomavirus (HPV) vaccination and effective precancer screening programs remain at the forefront, followed by treatment of any identified precancerous or cancerous lesions [4, 5]. In this regard, it is essential to distinguish between the concepts of cervical cancer elimination and HPV eradication. However, in pushing the overall agenda, the objective is to institute a target (tailored according to a country's existing health structures and economic circumstances), that, if reached, cervical cancer would no longer be considered a public health concern [5].

Although cytology-based cervical cancer screening programs have demonstrated usefulness in reducing attributable mortality rates in the developing world, they have been relatively more difficult to implement and scale up in developing countries like Ghana. This has been ascribed to lack of trained cytopathologists and pathology services and poor infrastructure for patient follow-up [6, 7]. As viable alternatives, visual inspection with dilute acetic acid (VIA) and HPV DNA testing have gained popularity in screening programs in low-resource settings. In such settings, the World Health Organization (WHO) conditionally recommends screen-and-treat algorithms by way of HPV testing and VIA [8]. These recommendations were based on the following pieces of low-quality evidence available on the accuracies of these methodologies: recommendation 2–preferably screening with VIA, followed by cryotherapy (or loop electrosurgical excision procedure [LEEP] if ineligible) *over* HPV screening followed by cryotherapy (or LEEP if ineligible); recommendation 6–preferably perform HPV testing followed by VIA and cryotherapy (or LEEP) *or* HPV testing with cryotherapy (or LEEP); and recommendation 7–preferably screen via HPV testing, followed by VIA with cryotherapy (or LEEP) *over* screening with VIA and treatment with cryotherapy (or LEEP) [8].

VIA is not expensive and can be performed by lay-trained healthcare professionals, and VIA-positive women can undergo treatment within the same visit, reducing the need to recontact patients and the resulting loss to follow-up. Despite these advantages, the performance of the procedure varies according to provider experience, with wide ranges of sensitivity (41–92%) and specificity (49–98%) for the detection of histologically-confirmed high-grade cervical precancerous lesions (CIN2+) [9, 10]. In addition, VIA findings are less reliable among menopausal patients [11]. In primary screening for precancerous lesions, HPV DNA testing has been shown to outperform cytology or VIA in decreasing cervical cancer incidence and mortality in a single screen [12, 13], with a sensitivity of 89.7% (range, 86.4–93.9) and specificity of 88.2% (range, 86.2–90.1) for detecting CIN2+ lesions [14]. However, women with transient high-risk HPV (hr-HPV) infection are at risk of overtreatment; thus, under ideal conditions, only women with progressive hr-HPV infection should be triaged for treatment [15].

At the Cervical Cancer Prevention and Training Centre (CCPTC), Battor, Ghana, primary screening via HPV DNA testing is employed using a number of platforms: AmpFire, GeneXpert, MA-6000, and previously *care*HPV. Visual inspection methods such as VIA and mobile colposcopy with the Enhanced Visual Assessment (EVA) system are options also available to clients. Various screening models of concurrent hr-HPV DNA technologies and visual inspection methods have yet to be evaluated for their general performance and comparative advantages and disadvantages specific to a low-resource setting. There is a gap in knowledge in this area of cervical cancer screening, particularly in low-resource settings. For a country with no

functional national cervical cancer screening guidelines, this study of concurrent HPV DNA and VIA testing in a secondary-level healthcare facility could potentially generate a wealth of information, that could shape the future of cervical cancer screening by formulating a strategy that works best for our setting.

The present study aimed to evaluate the existing recommendations of the WHO pertaining to visual inspection and HPV DNA testing algorithms for cervical cancer screening in a real-world low-resource setting. Specifically, we aimed to determine the detection rates of concurrent (combined) VIA or EVA mobile colposcopy and hr-HPV DNA testing in comparison to standalone hr-HPV DNA testing; to evaluate the rate of loss to follow-up in this cohort; and to discuss our approach to triaging hr-HPV-positive patients at our center.

## Materials and methods

### Ethical considerations

The Research Ethics Committee of the Catholic Hospital, Battor, granted ethical approval for this study (approval no. CHB-ERC-002/07/19). The need for informed consent was waived on account of the retrospective nature of the study.

### Study setting, design, and overview

The CCPTC was established in May 2017 and started training health workers in cervical cancer prevention skills (including how to perform VIA and colposcopy) in September 2017. Standalone HPV DNA testing was the main cervical precancer screening method. The algorithm used at the CCPTC for cervical precancer screening with HPV DNA testing has been published previously [16]. In November 2017, to enable trainees acquire more practical experience, VIA became a routine procedure at no extra cost to clients. It was at this time that the CCPTC started to gain experience with concurrent HPV DNA testing and visual inspection procedures using the algorithm presented in S1 Fig. Although VIA was performed at no additional cost, women had to pay from their pockets to undergo EVA mobile colposcopy and hr-HPV DNA testing. This made it possible to compare concurrent HPV DNA testing and visual inspection methods (VIA or mobile colposcopy) with standalone HPV DNA testing in a routine clinical setting. One hundred and thirty-five women who opted for and performed self-sampling were included in the standalone HPV DNA testing group as no visual inspection method was performed.

The present retrospective, descriptive cross-sectional study included all 4482 women subjected to cervical precancer screening via HPV DNA testing in combination with a visual inspection method (VIA or EVA colposcopy) and 1574 women subjected to standalone hr-HPV DNA testing at the CCPTC, Battor, between June 1, 2016 and March 31, 2022. All data on the women and their screening statuses were captured and stored securely in databases managed by the CCPTC. All personal data were de-identified prior to the analyses.

### Variables and outcomes

We captured and analyzed data pertaining to sociodemographic characteristics of the screened women, including age, marital status, parity, monthly income, education level, and religious faith. We also collected data regarding self-reported risk factors such as HIV and smoking status. The outcomes of interest were a positive hr-HPV DNA test determined using any of the four platforms and/or the presence of clinically relevant lesion(s) on visual inspection (using VIA or mobile colposcopy).

## Cervical sample collection and HPV DNA screening

During the screening visit, all women were counseled on the benefits of cervical screening and its associated risks and possible outcomes. In the dorsal lithotomy position, a speculum was placed to expose the cervix and to obtain cervical specimens with a sterile brush or dry cotton swab for laboratory processing and typing. Women who opted to undergo hr-HPV DNA testing with self-samples were instructed on Evalyn self-sample brush (Rovers Medical Devices B. V., Oss, Netherlands) use. The Evalyn brushes were capped after sample collection, stored in a dry area at room temperature, and returned to our central laboratory for processing within seven days.

HPV DNA testing was performed using *care*HPV (Qiagen GmBH, Hilden, Germany) [17], GeneXpert (Cepheid, Sunnyvale, CA, USA) [18], AmpFire (Atila BioSystems, Inc., Mountain View, CA, USA) [16, 19], or MA-6000 (Sansure Biotech Inc., Hunan, China) [20], depending on the period of testing. Each test was performed strictly according to the manufacturer's protocol.

## HPV DNA detection

*care*HPV utilizes a powerful and fast procedure to detect HPV DNA in cervical specimen collected into *care*HPV collection medium. The platform is designed to detect HPV 16/18/31/33/35/39/45/51/52/56/58/59/66/68 without distinction [17]. GeneXpert also detects HPV DNA in PreserCyt or ThinPrep liquid specimens. It specifically identifies HPV 16 and 18/45 and collectively identifies genotypes 31/33/35/39/51/52/56/58/59/66/68 as 'other' hr-HPV types [18]. Both AmpFire and MA-6000 detect HPV DNA in cervicovaginal samples (collected using dry brushes, swabs, or PreservCyt/ThinPrep) and specifically identify genotypes 16 and 18 and collectively identify genotypes 31/33/35/39/45/51/52/53/56/58/59/66/68 [19, 20]. Even though both AmpFire and MA-6000 offer full genotyping options, these are more expensive and have a low throughput, and so are not generally performed at our facility.

## Screening via visual inspection methods

VIA was performed by locally trained nurses under the supervision of a specialist gynecologist. The cervix was inspected after applying 5% acetic acid, waiting for 90–120 seconds, and looking over carefully for abnormal changes under a 100 W incandescent light source [7]. VIA positivity was defined as the presence of well-defined opaque aceto-whitening at the transformation zone.

Mobile colposcopy was performed using the EVA system (MobileODT, Tel Aviv, Israel), a platform built around a smartphone interface with an online image storage portal. The application is used to control the mobile colposcope and upload colposcopic images for review by a gynecologist [16, 21]. Nurses recorded details pertaining to colposcopic adequacy, the type of transformation zone, and the presence of any cervical or vaginal lesions.

Upon cervical screening by EVA colposcopy or VIA, the result of the examination was classified using the Rio 2011 Colposcopy Nomenclature of the International Federation for Cervical Pathology and Colposcopy (IFCPC) [22]. In keeping with this nomenclature, each woman's screening status was classified as one of the following:

1. Inadequate conditions for colposcopic assessment, inflammation

2. Adequate conditions for colposcopic assessment, transformation zone type 1

3. Adequate conditions for colposcopic assessment, transformation zone type 2

4. Adequate conditions for colposcopic assessment, transformation zone type 3; only limited assessment of the transformation zone is possible.

### Definitions of transformation zone types

Type 1: The entire circumference of the squamocolumnar junction is visible; fully ectocervical.

Type 2: The entire circumference of the squamocolumnar junction is visible; partly or fully endocervical.

Type 3: The entire circumference of the squamocolumnar junction is not visible; partly or fully endocervical.

### Statistical analysis

Descriptive statistics were generated for all sociodemographic and clinical variables. Percentages and counts were used to describe all categorical variables and prevalence estimates, alongside their binomial exact 95% confidence intervals (CIs). Continuous variables are described as means with their standard deviations (SDs) or medians with their interquartile ranges, depending on the level of skewness. Overall rates of positivity on EVA colposcopy, VIA, and hr-HPV testing are presented and disaggregated by HIV status and test platform. All statistical analyses were performed using Stata 15 (StataCorp LLC, College Station, TX, USA).

## Results

### Sociodemographic and clinical details of participants

A total of 4482 women underwent concurrent cervical precancer screening via HPV DNA testing and visual inspection (VIA or EVA colposcopy) during the study period. The sociodemographic, clinical, and screening characteristics of the study participants are presented in Table 1. The mean age at screening was 39.3 (SD, 9.4) years, with a majority of women being married (51%) or having a steady partner (22%). Almost 1 in 4 (23.1%) of the women screened had completed tertiary education and almost 1 in 10 (10.8) had no formal education. As self-reported risk factors, most participants had never smoked (99.5%) and were HIV negative (54%). The HIV positivity rate among 2538 women with available information on HIV status was 5.0%.

### Overall and subgroup analyses of hr-HPV DNA detection rates

Overall, at screening, 800 of 4482 participants (17.9%; 95% CI, 16.7–19.0) tested hr-HPV positive, whereas 69 of 801 (8.6%; 95% CI, 6.7–10.6) women who underwent EVA colposcopy were 'positive' and 76 out of 3681 women (2.1%; 95% CI, 1.6–2.5) who underwent VIA were 'positive'. The corresponding rates tended to be higher among HIV-positive women (10.5%, 7.5%, and 42.5% positivity rates on EVA colposcopy, VIA, and hr-HPV DNA testing, respectively) (Table 1). When disaggregated according to test platform, MA-6000 showed the highest hr-HPV detection rate of 24.6% (95% CI, 21.6–27.7), followed by GeneXpert (19.0%; 95% CI, 12.6–25.5), AmpFire (16.6%; 95% CI, 15.3–17.9), and *care*HPV (13.4%; 95% CI, 9.6–17.2).

### Overall screening outcomes and treatment of study participants

In the subgroup of 3681 women who underwent VIA, 3106 were hr-HPV negative, among whom 55 (1.8%) were VIA 'positive' (Fig 1). Among 575 hr-HPV-positive women (15.6%) in the same subgroup, 21 (3.7%) were VIA 'positive'. Eight of these women were treated based on hr-HPV-negative and VIA-positive results: thermal coagulation in 3 and LEEP in 5. Two women were treated based on hr-HPV-positive and VIA-positive results: thermal coagulation and LEEP

**Table 1. Sociodemographic and clinical characteristics of women (n = 4482) who underwent concurrent cervical precancer screening via HPV DNA testing and visual inspection.**

| Characteristic | Estimate |
|---|---|
| **Age, mean (SD)** | 39.3 (9.4) |
| **Marital status, n (%)** | |
| Single | 667 (14.9) |
| Has a steady partner | 987 (22.0) |
| Married | 2295 (51.2) |
| Divorced | 307 (6.9) |
| Widowed | 206 (4.6) |
| Missing | 20 (0.5) |
| **Number of children, median (IQR)** | 1 (0, 2) |
| **Highest level of education, n (%)** | |
| No formal education | 483 (10.8) |
| Elementary education | 842 (18.8) |
| Secondary education | 2006 (44.8) |
| Tertiary education | 1034 (23.1) |
| Vocational/technical/other | 114 (2.5) |
| Missing | 3 (0.1) |
| **Religious faith, n (%)** | |
| Christian | 3998 (89.2) |
| Islam | 269 (6.0) |
| African traditional religion | 22 (0.5) |
| Other | 7 (0.2) |
| None | 5 (0.1) |
| Missing | 181 (4.0) |
| **Smoker, n (%)** | 20 (0.5) |
| **HIV status, n (%)** | |
| Positive | 127 (2.8) |
| Negative | 2411 (53.8) |
| Unknown | 1944 (43.4) |
| **Earns income, n (%)** | |
| Yes | 3851 (85.9) |
| No | 440 (9.8) |
| Missing | 191 (4.2) |
| **EVA positive, % (95% CI)** | 8.6 (6.7–10.6) |
| EVA positive, % (95% CI) [among 86 HIV positive women] | 10.5 (4.0–16.9) |
| **VIA positive, % (95% CI)** | 2.1 (1.6–2.5) |
| VIA positive, % (95% CI) [among 41 HIV positive women] | 7.3 (0.0–15.3) |
| **hr-HPV positive, % (95% CI)** | 17.9 (16.7–19.0) |
| hr-HPV positive, % (95% CI) [among HIV 127 positive women] | 42.5 (33.9–51.1) |
| **hr-HPV positive by test platform, % (95% CI)** | |
| *care*HPV | 13.4 (9.6–17.2) |
| GeneXpert | 19.0 (12.6–25.5) |
| AmpFire | 16.6 (15.3–17.9) |
| MA-6000 | 24.6 (21.6–27.7) |

hr-HPV, high-risk human papillomavirus

SD, standard deviation

IQR, interquartile range

VIA, visual inspection with dilute acetic acid

CI, confidence interval.

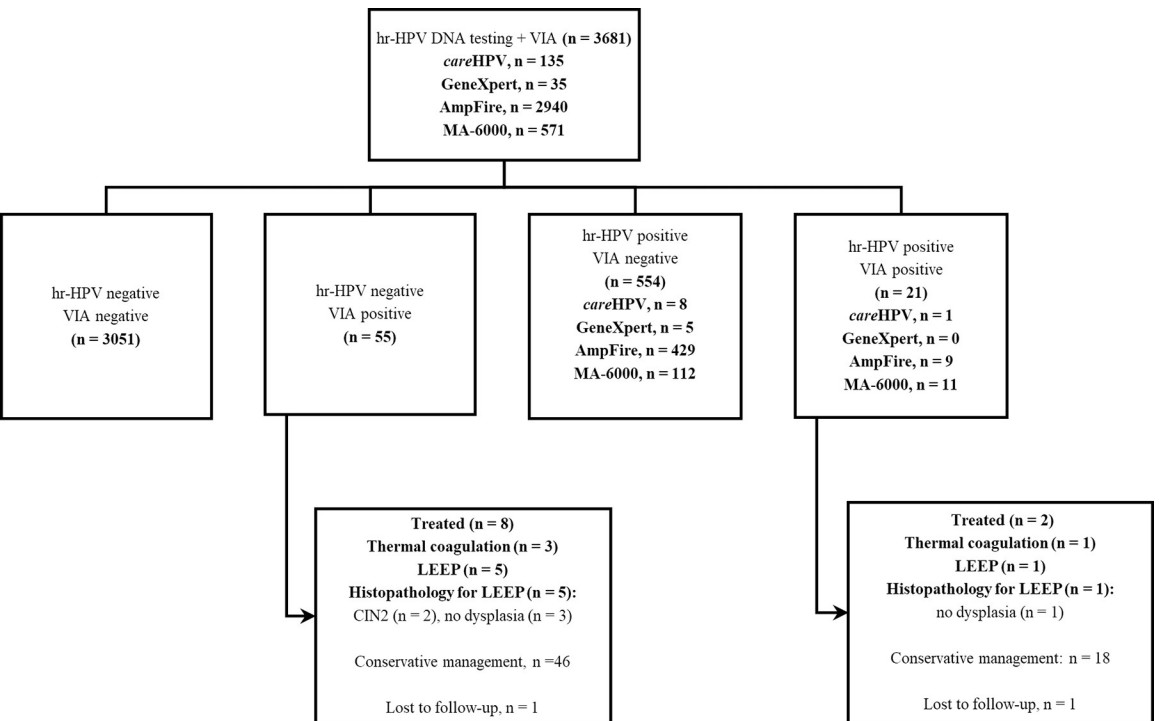

**Fig 1. Flow chart for concurrent screening with hr-HPV DNA testing and VIA.** VIA, visual inspection with dilute acetic acid; hr-HPV, high-risk human papillomavirus; LEEP, loop electrosurgical excision procedure.

in 1 each. In the hr-HPV-negative VIA positive group, histopathology following LEEP revealed no dysplasia in 3 women and CIN2 in 2 women. In the hr-HPV-positive VIA positive group, histopathology revealed no dysplasia in the only woman subjected to LEEP (Fig 1).

Within the subgroup of 801 (17.9%) women subjected to concurrent hr-HPV DNA testing and EVA colposcopy, 39 (4.9%) showed an abnormal lesion on EVA colposcopy but tested hr-HPV negative, whereas 195 (24.3%) tested hr-HPV positive but EVA 'negative' and 30 (3.7%) tested hr-HPV positive and EVA 'positive' (Fig 2). Six out of 39 women who tested hr-HPV negative and EVA positive were treated: three each by way of thermal coagulation or LEEP. Histopathology following LEEP for these 3 women showed no dysplasia. Seventeen out of 30 women were treated based on hr-HPV positivity and EVA positivity: thermal coagulation in 6 and LEEP in 11. Histopathology following LEEP revealed CIN2 lesions in 5 women, CIN3 lesions in 2 women, and no dysplasia in the remaining four women (Fig 2).

## Outcomes of women subjected to concurrent hr-HPV testing and visual inspection by test combination and follow-up after standalone hr-HPV testing

Table 2 shows the distributions of screening outcomes stratified according to method and platform combinations. The highest rate of positivity on both hr-HPV DNA testing and visual inspection was recorded for GeneXpert +EVA (4.7%; 95% CI, 1.5–10.6), followed by AmpFire + EVA (4.5%; 95% CI, 2.5–7.4), *care*HPV + EVA (3.9%; 95% CI, 1.6–7.9), and MA-6000 + EVA (2.5%; 95% CI, 0.8–5.7). Overall, 51 women in the entire cohort showed positive results on both hr-HPV DNA testing and a visual inspection method (1.1%; 95% CI, 0.9–1.5), whereas a large majority (80.0%; 95% CI, 78.8–81.2) of the women showed negative findings on both

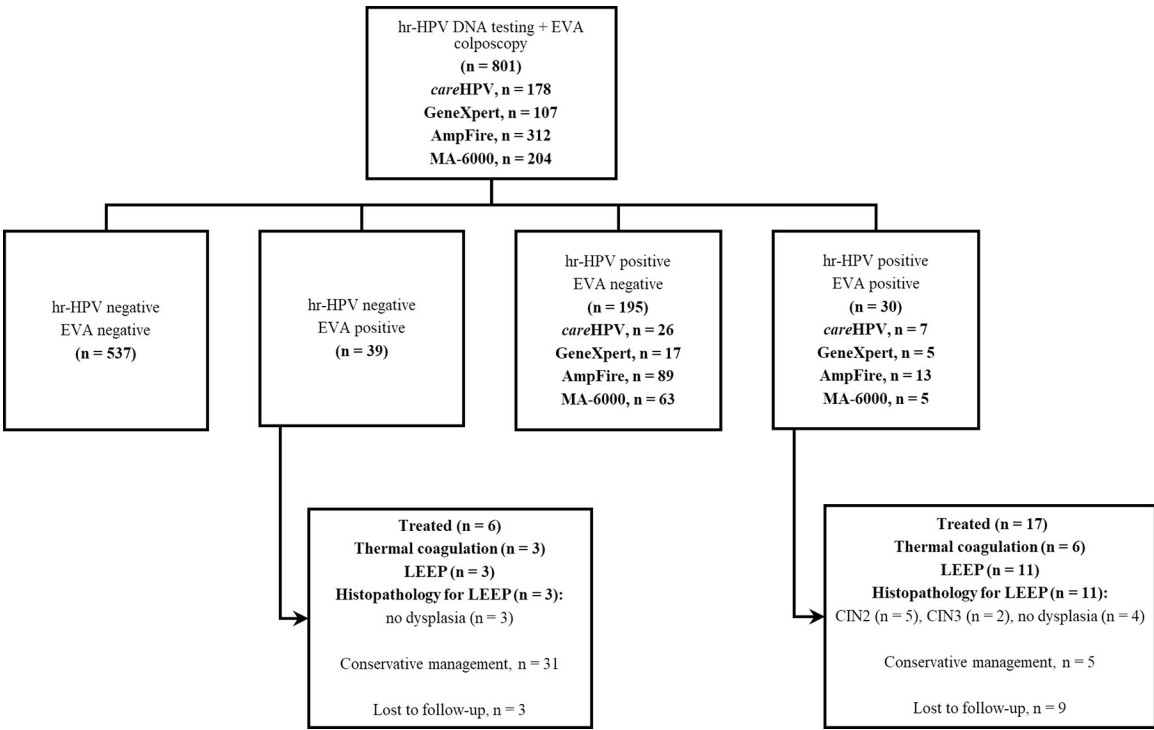

**Fig 2. Flow chart for concurrent screening with hr-HPV DNA testing and EVA colposcopy.** hr-HPV, high-risk human papillomavirus; LEEP, loop electrosurgical excision procedure.

hr-HPV testing and visual inspection. Again, 191 (69.5%) of 275 women who tested hr-HPV positive on any platform applied, as a standalone test for screening, returned for at least one follow-up visit within the study period (Table 3).

## Discussion

Our study aimed to evaluate the detection rates of concurrent hr-HPV DNA testing and visual inspection by way of EVA mobile colposcopy or VIA in comparison to standalone hr-HPV DNA

**Table 2. Distributions of results of concurrent hr-HPV testing and visual inspection screening stratified by method.**

| | hr-HPV + | hr-HPV + | hr-HPV – | hr-HPV – | Total |
|---|---|---|---|---|---|
| | Visual inspection – | Visual inspection + | Visual inspection + | Visual inspection – | |
| *care*HPV + VIA, n (%) | 8 (5.9) | 1 (0.7) | 3 (2.2) | 123 (91.1) | 135 |
| *care*HPV + EVA, n (%) | 26 (14.6) | 7 (3.9) | 10 (5.6) | 135 (75.8) | 178 |
| GeneXpert + VIA, n (%) | 5 (14.3) | 0 (0.0) | 0 (0.0) | 30 (85.7) | 35 |
| GeneXpert + EVA, n (%) | 17 (15.9) | 5 (4.7) | 6 (5.6) | 79 (73.8) | 107 |
| AmpFire + VIA, n (%) | 429 (14.6) | 9 (0.3) | 36 (1.2) | 2466 (83.9) | 2940 |
| AmpFire + EVA, n (%) | 89 (28.5) | 13 (4.2) | 13 (4.2) | 197 (63.1) | 312 |
| MA-6000 + VIA, n (%) | 112 (19.6) | 11 (1.9) | 16 (2.8) | 432 (75.7) | 571 |
| MA-6000 + EVA, n (%) | 63 (30.9) | 5 (2.5) | 10 (4.9) | 126 (61.8) | 204 |
| Total, n (%) | 749 (16.7) | 51 (1.1) | 94 (2.1) | 3588 (80.1) | 4482 |

VIA, visual inspection with dilute acetic acid

hr-HPV

high-risk human papillomavirus.

**Table 3. Distributions of hr-HPV test results and follow-up rates following standalone hr-HPV testing stratified by platform.**

|  | hr-HPV + | hr-HPV − | hr-HPV + cases who returned for follow-up |
|---|---|---|---|
| *care*HPV, n (%) | 166 (15.1) | 933 (84.9) | 104 (62.7) |
| GeneXpert, n (%) | 28 (11.6) | 213 (88.4) | 25 (89.3) |
| AmpFire, n (%) | 39 (31.7) | 84 (68.3) | 27 (69.2) |
| MA-6000, n (%) | 42 (37.8) | 69 (62.2) | 35 (83.3) |
| Total, n (%) | 275 (17.5) | 1299 (82.5) | 191 (69.5) |

testing among 4482 women who presented for cervical precancer screening at the CCPTC, Battor. This work will provide information and more options to the WHO recommendation of a 'see and treat' approach to cervical cancer screening in low-resource settings because it is cost-effective and reduces loss to follow-up [7, 8, 23]. To the best of our knowledge, the present work is the first to document an evaluation of concurrent cervical precancer screening as an alternative to the see and treat approach advised by the WHO for resource-limited countries.

One of the most important contributions of the study was to evaluate retention in care, perhaps one of the greatest challenges in cervical cancer screening programs worldwide. When standalone hr-HPV DNA testing was performed, 69.5% (191/275) of hr-HPV-positive women returned for follow-up evaluation via visual inspection (colposcopy), despite various efforts made to reach all of them. This finding demonstrates that without performing concurrent visual inspection (colposcopy or VIA) within the same visit as when a cervicovaginal sample was taken for hr-HPV testing, as many as 30.5% of hr-HPV-positive women would be lost to follow-up. This high rate of loss to follow-up is multifactorial, and despite exerting reasonable efforts to reduce loss to follow-up, some barriers remained that might best be elucidated through qualitative research. Ghana remains a vibrant mobile phone market in the sub-Saharan region, with recent surveys showing that 83% of all adults own a mobile phone and that there are approximately 130 mobile service subscribers per 100 population [24, 25]. Although the country has a high penetration of mobile phone services, it may be difficult to get hr-HPV-positive women to screening centers on multiple occasions due to additional costs associated with transportation. Further, even though attempts have been made in recent times to implement a digital addressing system for houses in many cities and towns in Ghana in addition to street naming [26], many Ghanaians do not have permanent addresses, making a call-recall system for a national cervical cancer prevention program difficult to implement. Further, in similar low-resource settings, a lack of understanding of cervical cancer as a disease, the screening process, and its outcomes, linked with poor socioeconomic circumstances and low levels of male partner support have been identified as hampering adherence to follow-up after an initial positive screening test [27, 28]. Thus, besides national policies, at the facility level, it may be useful to intensify information-education-communication activities, in addition to incentivizing providers and screen-positive attendees to complete the continuum of cancer care. All these factors contribute to loss to follow-up, and make a case for a single visit approach to cervical cancer screening in Ghana and other low (middle) income countries. The training program offered at the CCPTC for nurses and midwives across Ghana and beyond provides most women screened with hr-HPV DNA testing the opportunity to undergo VIA in a single visit and in the same setting (when the HPV sample was taken). In addition, it allows trainees to have hands-on experience in performing VIA, while giving attendees the opportunity to have a one-time visit without paying additional fees for VIA. In light of the foregoing, we posit that concurrent testing, where available, may be more cost-effective than waiting for hr-HPV DNA results to recall hr-HPV-positive women for colposcopy.

Another considerable challenge was highlighted in our data: namely, 2.1% (94/4482) of our study participants showed negative results on hr-HPV testing but 'positive' results on visual inspection (colposcopy or VIA). The corresponding rates were 4.9% (39/801) for the EVA colposcopy group and 1.5% (55/3681) for the VIA group. These women represent a very important group because these data show that if hr-HPV DNA testing had been performed as a standalone test, and only hr-HPV-positive women had been recalled for colposcopy/VIA, potentially precancerous cervical lesions would have been missed. This proportion would further be expected to be greater among high-risk groups such as women living with HIV [29]. In total, 14.9% (14/94) of these women with cervical lesions underwent treatment: 6 in the EVA colposcopy group and 8 in the VIA group. In this group of hr-HPV-negative visual inspection 'positive' women, 6 were treated with thermal coagulation (thus, no tissue was obtained for histopathology) while 8 were treated with LEEP. The histopathology reports for the 8 women who underwent LEEP were no dysplasia (n = 6) and CIN2 (n = 2). Many factors determined who got treated and the type of treatment (ablation–thermal coagulation or excisional–LEEP). LEEP was recommended for large lesions that cover >75% of the ectocervix or extend into the endocervical canal. There was also the option of conservative management (follow-up) for women with minor changes on colposcopy and who were unlikely to be lost to follow-up, while prompt treatment was recommended for women with major changes. In the 2011 Colposcopy Nomenclature of the IFCPC, minor changes (Grade 1) refer to thin acetowhite epithelium, irregular, geographic border, fine mosaic, and fine punctuation. Major changes (Grade 2) refer to dense acetowhite epithelium, rapid appearance of aceto-whitening, cuffed gland openings, coarse mosaic, coarse punctuation, sharp border, and the presence of an inner border sign or ridge sign [30]. Representative colposcopic images obtained from women who tested EVA 'positive' and hr-HPV DNA positive on the *care*HPV, GeneXpert, AmpFire, and MA-6000 platforms are shown in S2–S5 Figs.

Further, we observed a disparity between the positivity rates for VIA and EVA colposcopy. While the VIA 'positivity' rate was 2.1 (95% CI, 1.6–2.5), the EVA colposcopy 'positivity' rate was 8.6% (95% CI, 6.7–10.6), representing a clear difference of 6.5% (95% CI, 4.6–8.5; $p<0.0001$). In addition to the established relatively more operator-dependent nature of VIA [7, 10, 17, 29], this observed difference may be attributable to the magnification offered by the mobile colposcope, rendering small lesions not visible to the naked eye on VIA visible during colposcopy.

## Strengths and limitations

The main strengths of this study lay in its relatively large sample size, allowing us to stratify prevalence and detection rates without loss of statistical power, as well as its novelty as the first study, to the best of our knowledge, to evaluate a concurrent cervical precancer screening algorithm in Ghana. However, our study had a number of limitations, from which future studies can draw to improve generalizability to similar low-resource settings. First, this work was done in a routine clinical setting at Catholic Hospital, Battor. As this was not a funded project, the women paid from their pockets to get screened and treated. When offered the choice of treatment approach (thermal coagulation or LEEP), many, who could not afford biopsies, LEEP, and the cost of histopathology opted to undergo ablation. When thermal coagulation was chosen, no tissue was obtained for histopathology. Thus, we were unable to present the histopathological results for many of the cervical lesions seen and followed up or treated by ablation. Again, because we were unable to perform full genotyping for hr-HPV positive women, we could neither distinguish among recognized, probable (HPV 66 and 68), or potential (HPV 53) hr-HPV genotypes nor account for the bias that could have arisen from using multiple

testing platforms. In addition, the criterion for adjudging positivity on VIA or mobile colposcopy is commonly the presence of aceto-whitening. Aceto-whitening may be due to immature metaplasia, inflammation, subclinical papillomavirus infection, or CIN. Biopsies for histopathology would have been useful to confirm precancerous (CIN2+) lesions. This work could also not address the proportion of hr-HPV negatives who were visual inspection method (VIA/colposcopy) negative but had lesions in the endocervical canal, that is for women with transformation zone type 3. This is because pap smears and/or endocervical curettage were not taken for this group of women. Further studies are needed to identify the exact proportion of precancerous lesions that are hr-HPV DNA negative but positive on visual inspection or negative on visual inspection because the lesions are in the endocervical canal. Evidence suggests that this group may not constitute a clinically important group if clinically validated HPV assays are used. However, we wish to create awareness among health workers in low-resource settings that HPV assays available (such as those used in our study) may be less sensitive than the gold standard and other validated assays. Thus, the increasing popularity of such platforms in low-resource settings makes this group increasingly important. Again, our findings on the level of education of these women suggest that they were more likely to have a tertiary level of education (23.1%) than the general female population (10.4%) of Ghana according to the 2021 Population and Housing Census [31]. This suggests that the results of our research may not be generalizable to the entire female population of Ghana. Last, the completeness of medical records and missing data represented minor challenges. To mitigate these, we reviewed all data sources in-depth to minimize the effects of missingness on our estimates and included 'missing data' as an outcome measure for all relevant variables.

## Conclusion

Based on our study population, prevailing poor socioeconomic circumstances, additional transportation costs associated with multiple screening visits, and lack of a reliable address system in many parts of Ghana, we posit that standalone HPV DNA testing with recall of hr-HPV positives will be tedious for a national cervical cancer prevention program. Our preliminary data show that concurrent testing (hr-HPV DNA testing along with visual inspection by way of VIA or mobile colposcopy) may be more cost-effective than recalling hr-HPV-positive women for colposcopy. In addition to building significant levels of capacity and strengthening existing health systems, a concurrent approach, where available, may better address the entire continuum of cervical cancer screening and improve loss to follow-up in low-resource countries.

## Supporting information

**S1 Fig. Algorithm for screening with concurrent HPV DNA testing and visual inspection at the CCPTC.**
(TIF)

**S2 Fig. Colposcopy images of a 41-year-old woman, para 2.** *care*HPV testing was performed concurrently with EVA colposcopy: (A) before applying acetic acid and (B) after applying acetic acid. *care*HPV–positive; EVA transformation zone type 3, dense aceto-whitening more anteriorly; treatment–LEEP; histopathology, CIN2.
(TIF)

**S3 Fig. Colposcopy images of a 37-year-old woman, para 2.** GeneXpert testing was performed concurrently with EVA colposcopy: (A) before applying acetic acid and (B) after applying acetic acid. GeneXpert–positive (others, P3); EVA transformation zone type 3,

circumferential aceto-whitening, dense at the 1–3 o'clock position; treatment–LEEP; histopathology, CIN2.
(TIF)

**S4 Fig. Colposcopy images of a 37-year-old woman, para 0.** AmpFire hr-HPV testing was performed concurrently with EVA mobile colposcopy: (A) before applying acetic acid and (B) after applying acetic acid. AmpFire–positive for HPV 18; EVA–leukoplakia with circumferential dense aceto-whitening; treatment–LEEP; histopathology, CIN 3.
(TIF)

**S5 Fig. Colposcopy images of a 26-year-old woman, para 0+1.** MA-6000 HPV DNA testing was performed concurrently with EVA colposcopy. MA-6000–positive for 'other' HPV type (s); EVA–adequate, transformation zone type 1, thin aceto-whitening on the anterior and posterior cervical lips; treatment–thermal coagulation.
(TIF)

## Acknowledgments

The authors acknowledge all the workers in the main laboratory of Catholic Hospital, Battor, and the laboratory at the CCPTC as well as the staff at the CCPTC and the Department of Obstetrics and Gynecology, Catholic Hospital, Battor who contributed in various ways toward the screening and management of these women. The authors also thank the Catholic Hospital, Battor for its support.

## Author Contributions

**Conceptualization:** Kofi Effah, Ethel Tekpor, Comfort Mawusi Wormenor, Joseph Emmanuel Amuah, Patrick Kafui Akakpo.

**Data curation:** Kofi Effah, Ethel Tekpor, Comfort Mawusi Wormenor, Joseph Emmanuel Amuah, Nana Owusu Essel, Bernard Hayford Atuguba, Gifty Belinda Klutsey, Edna Sesenu, Georgina Tay, Faustina Tibu, Seyram Kemawor, Isaac Gedzah, Esu Aku Catherine Morkli, Stephen Danyo.

**Formal analysis:** Kofi Effah, Ethel Tekpor, Comfort Mawusi Wormenor, Joseph Emmanuel Amuah, Nana Owusu Essel, Esu Aku Catherine Morkli, Stephen Danyo.

**Investigation:** Kofi Effah, Ethel Tekpor, Gifty Belinda Klutsey, Edna Sesenu, Georgina Tay, Faustina Tibu, Seyram Kemawor, Isaac Gedzah, Stephen Danyo.

**Methodology:** Ethel Tekpor, Joseph Emmanuel Amuah.

**Project administration:** Kofi Effah.

**Resources:** Kofi Effah.

**Software:** Joseph Emmanuel Amuah.

**Supervision:** Kofi Effah.

**Validation:** Kofi Effah.

**Writing – original draft:** Kofi Effah, Ethel Tekpor, Comfort Mawusi Wormenor, Joseph Emmanuel Amuah, Nana Owusu Essel, Patrick Kafui Akakpo.

**Writing – review & editing:** Kofi Effah, Ethel Tekpor, Comfort Mawusi Wormenor, Joseph Emmanuel Amuah, Nana Owusu Essel, Patrick Kafui Akakpo.

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
