## [Decision Letter · Decision Letter 0]

31 Jan 2023

PGPH-D-22-02125

Concurrent HPV DNA testing and a visual inspection method for cervical precancer screening: a practical approach from Battor, Ghana

Dear Dr. Essel,

Thank you for submitting your manuscript to PLOS Global Public Health. After careful consideration, we feel that it has merit but does not fully meet PLOS Global Public Health’s publication criteria as it currently stands. Therefore, we invite you to submit a revised version of the manuscript that addresses the points raised during the review process.

EDITOR:

In revising your manuscript, kindly take careful consideration of the detailed comments provided by the reviewersPlease take note of the suggested edits in the annotated pdf which may be useful in finalizing your revised manuscript

We look forward to receiving your revised manuscript.

Kind regards,

Edina Amponsah-Dacosta, Ph.D., MPH

Academic Editor

Journal Requirements:

Additional Editor Comments (if provided):

Reviewers' comments:

Reviewer's Responses to Questions

**Comments to the Author**

1. Does this manuscript meet PLOS Global Public Health’s publication criteria? Is the manuscript technically sound, and do the data support the conclusions? The manuscript must describe methodologically and ethically rigorous research with conclusions that are appropriately drawn based on the data presented.

Reviewer #1: Yes

Reviewer #2: Partly

Reviewer #3: Yes

2. Has the statistical analysis been performed appropriately and rigorously?

Reviewer #1: Yes

Reviewer #2: No

Reviewer #3: Yes

3. Have the authors made all data underlying the findings in their manuscript fully available (please refer to the Data Availability Statement at the start of the manuscript PDF file)?

Reviewer #1: Yes

Reviewer #2: Yes

Reviewer #3: Yes

4. Is the manuscript presented in an intelligible fashion and written in standard English?

Reviewer #1: Yes

Reviewer #2: Yes

Reviewer #3: Yes

5. Review Comments to the Author

Reviewer #1: The manuscript describes the results of a descriptive cross-sectional study on cervical cancer prevention conducted in Ghana, . Overall, during June 2016 through December 2022, 4483 women (median age 39.3 yrs) underwent concurrent HPV DNA testing (by means of different assays, depending on the period) and visual inspection (VIA or EVA - mobile colposcopy); in particular, a subgroup of 3681 women by VIA, and a subgroup of 802 women by EVA; the women had to pay from their pockets for HPV testing, EVA and treatment (only VIA was performed at no extra charge. Moreover, a subgroup of 1299 women underwent a stand-alone HPV DNA test on a self-collected sample. The rates of positive results were markedly different between the three approaches; 17.9% for HPV DNA testing, 2.0% for VIA, and 8.8% for EVA. A concordant positive result was observed in 1.2% and a concordant negative result in 80% of the women. Overall, 33 women underwent treatment, by either thermal coagulation (n=13) or LEEP (n=20); as a result, a histologic diagnosis is available only for a very small number of women. The aims of the study were (1) to gain data on how cervical cancer screening may be effectively implemented in Ghana (and other low income countries), and (2) whether and how a "see and treat" approach can be safely and efficiently used in these settings. The topic of cervical cancer prevention in low income countries is very important because in these areas cervical cancer is highly frequent and prevention by cytology is very difficult to implement. Since over 30% (84/276) of the women with a positive stand-alone HPV test did not go back for follow up, the authors conclude that a concurrent approach may be more cost-effective than recalling HPV-positive women for colposcopy.

MAJOR COMMENTS:

1-METHODS: a more precise description of when the visual inspection and the treatments were performed, as well as more information on the subgroup of women who underwent stand-alone HPV testing (i.e., all by self-sampling?), should be added.

2-PARTICIPANTS: the characteristics reported in Table 1 on the number of children, the degree of education and the income earning identify these women as a selected group; this should be stated and the results commented also in relation to the generalizability of the findings.

3-HIV INFECTION (page 8, line 190): the 54% HIV-negative fraction indicated does not take into account the fact that the HIV status was known for only slightly more than half of the included women, and should be better expressed as % of 2540 (that makes a 5% HIV positivity). Moreover, it should be interesting to separately report the outcomes in the 128 HIV-positive women.

4-DISCORDANT RESULTS: the HPV positivity rate was much higher than the positivity in visual inspection; the authors express concern for the risk of missing lesions in case of adoption of a stand-alone HPV testing; indeed, to this regard a very large amount of data in both high- and low-income countries exist demonstrating that HPV testing is more sensitive than cytology in detecting clinically relevant lesions (particularly CIN3+), although less specific; moreover, visual inspection has been shown to have a low specificity. Although a very small number of histological diagnoses have been performed in this study, the results (no dysplasia in 6/8 and CIN2 in 2/8 women HPV-negative & visual inspection positive) are in line with those data. The hrHPV-negative group, granted the use of clinically validated HPV assays, is not a clinically important group (as reported in page 17, line 353).

Reviewer #2: The abstract summarizes the study findings and clearly highlights the context and problem that authors wish to address.

The authors report on a practical approach of using two methods for cervical precancer screening. They included a large sample (4438 women) and compared the rates of positivity between the two screening approaches (a) combined testing using HPV DNA testing and Visual inspection; and (b) stand-alone high risk HPV testing.

The study presents the results of primary scientific research and their preliminary data supports concurrent testing as cost-effective, reducing loss to follow-up compared to recalling high risk HPV positive women for colposcopy (this is in agreement with recent literature). The study design and conclusions are sound, presented in an appropriate fashion and supported by the data - with the main limitation being the loss of histopathological results for many of the cervical lesions seen and followed up or treated by ablation, which made it impossible for the authors to confirm precancerous lesions in this case. This is generally a good study with minor suggestions provided below:

1. It is not clear from the data analyses and/or results how the four different high risk HPV tests compare to each other as this may have implications on the overall HPV prevalence reported in the study, especially if one test is more sensitive than the other - Additionally, it would have been interesting to see the high risk HPV positivity rate of all samples if they were all screened using a gold standard method in comparison to the 4 tests included in the study.

2. This is a retrospective study that used a descriptive cross-sectional study design, however, the authors only used descriptive statistics to analyse their data. The reviewer suggests considering also using confidence intervals (CI) not just on odds ratio estimates, because point estimates alone do not tell a meaningful story, instead, include a 2-by-2 table for comparing the two screening approaches and perhaps also further include univariable and multivariable logistic regression models to adjust for any potential confounding variables.

Finally, great work on producing such a well-written and interesting paper.

Reviewer #3: Thank you for the submission of your work to the journal. Congratulations on the great work done in your country towards the elimination of cervical cancer.

Since the authors used retrospective approach to the study there was a number of loss to follow up in the system, maybe if this was a prospective study design and patients knew that they were part of the study the loss to follow up could have better however this will mask this real life situation in the screening and treatment setting.

In the results the authors mention risk factors however these were not mentioned in the methods and how they were collected and which of the data was retrieved from the system for analysis. The self sampling method used is not well explained in the methods, although self sampling method have been found to be as sensitive as the doctor collected samples, the self sampling devices differ and therefore this may bring bias in the results.

This study outlines difficulties faced in low resource settings for cervical cancer screening and other screening programs. If women have to pay for their treatment the treatment is then bias and only those who can afford will receive treatment.

The issue of participants testing positive using VIA or EVA and testing negative with HPV could be due to the sensitivity of the test and the gene the test targets, its unfortunate that in the setting different HPV tests were employed and thus does not give a clear conclusion on HPV testing alone, furthermore this creates bias.

The VIA is subject to the user and therefor results may not be consistent throughout users. Then again the VIA was conducted by the nurses while the colposcopy was conducted by the gynecologist (highly qualified and expert in the field).

The limitations of the study are well outlined and explained by the authors however the conclusion is too long and the authors are discussing results instead of concluding based on the results.

Lastly the S1 figure is already published elsewhere as it is.

I have a concern though with publication of the images from colposcopy since this was a retrospective study the authors did not seek consent from the patients, however i am not sure if they had consent from the hospital or clinic to publish the images.

6. PLOS authors have the option to publish the peer review history of their article (what does this mean?). If published, this will include your full peer review and any attached files.

**Do you want your identity to be public for this peer review?** For information about this choice, including consent withdrawal, please see our Privacy Policy.

Reviewer #1: No

Reviewer #2: No

Reviewer #3: No

---

## [Decision Letter · Decision Letter 1]

28 Mar 2023

Concurrent HPV DNA testing and a visual inspection method for cervical precancer screening: a practical approach from Battor, Ghana

PGPH-D-22-02125R1

Dear Dr Essel,

We are pleased to inform you that your manuscript 'Concurrent HPV DNA testing and a visual inspection method for cervical precancer screening: a practical approach from Battor, Ghana' has been provisionally accepted for publication in PLOS Global Public Health.

Best regards,

Edina Amponsah-Dacosta, Ph.D., MPH

Academic Editor

Reviewer Comments (if any, and for reference):

Reviewer's Responses to Questions

**Comments to the Author**

1. If the authors have adequately addressed your comments raised in a previous round of review and you feel that this manuscript is now acceptable for publication, you may indicate that here to bypass the “Comments to the Author” section, enter your conflict of interest statement in the “Confidential to Editor” section, and submit your "Accept" recommendation.

Reviewer #1: All comments have been addressed

Reviewer #2: All comments have been addressed

Reviewer #3: All comments have been addressed

2. Does this manuscript meet PLOS Global Public Health’s publication criteria? Is the manuscript technically sound, and do the data support the conclusions? The manuscript must describe methodologically and ethically rigorous research with conclusions that are appropriately drawn based on the data presented.

Reviewer #1: (No Response)

Reviewer #2: Yes

Reviewer #3: Yes

3. Has the statistical analysis been performed appropriately and rigorously?

Reviewer #1: (No Response)

Reviewer #2: Yes

Reviewer #3: Yes

4. Have the authors made all data underlying the findings in their manuscript fully available (please refer to the Data Availability Statement at the start of the manuscript PDF file)?

Reviewer #1: (No Response)

Reviewer #2: Yes

Reviewer #3: Yes

5. Is the manuscript presented in an intelligible fashion and written in standard English?

Reviewer #1: (No Response)

Reviewer #2: Yes

Reviewer #3: Yes

6. Review Comments to the Author

Reviewer #1: (No Response)

Reviewer #2: Thank you for your revised manuscript submission.

1. The four different hr-HPV tests used in this study have been well explained, with differences and limitations of each platform clearly highlighted (lines 162-172).

2. In order to adjust for any potential confounding variables, the authors have presented and disaggregated the results for the different plartforms used in the study (EVA colposcopy, VIA, and hr-HPV testing) by self-reported risk factors such as HIV status (lines 207-208), as shown in Table1 and lines 263-275

Reviewer #3: Dear Authors

Thank you for the diligence in responding to the comments and suggestions made to the paper submitted. I am satisfied with the implementation of the suggestions and the corrections made to the paper.

Good luck with your publication.

7. PLOS authors have the option to publish the peer review history of their article (what does this mean?). If published, this will include your full peer review and any attached files.

**Do you want your identity to be public for this peer review?** For information about this choice, including consent withdrawal, please see our Privacy Policy.

Reviewer #1: No

Reviewer #2: No

Reviewer #3: No
